# Antidepressant Advisor (ADeSS): a decision support system for antidepressant treatment for depression in UK primary care – a feasibility study

Phillippa Harrison ,[1] Ewan Carr ,[2] Kimberley Goldsmith,[2] Allan Young,[1,3] Mark Ashworth ,[4] Diede Fennema,[1] Suqian Duan,[1] Barbara M Barrett,[5] Roland Zahn [1,3]

For numbered affiliations see end of article.

**Correspondence to**
Professor Roland Zahn;
Roland.zahn@kcl.ac.uk

## ABSTRACT

**Objectives** To develop and probe the first computerised decision-support tool to provide antidepressant treatment guidance to general practitioners (GPs) in UK primary care.
**Design** A parallel group, cluster-randomised controlled feasibility trial, where individual participants were blind to treatment allocation.
**Setting** South London NHS GP practices.
**Participants** Ten practices and eighteen patients with treatment-resistant current major depressive disorder.
**Interventions** Practices were randomised to two treatment arms: (a) treatment-as-usual, (b) computerised decision support tool.
**Results** Ten GP practices participated in the trial, which was within our target range (8–20). However, practice and patient recruitment were slower than anticipated and only 18 of 86 intended patients were recruited. This was due to fewer than expected patients being eligible for the study, as well as disruption resulting from the COVID-19 pandemic. Only one patient was lost to follow-up. There were no serious or medically important adverse events during the trial. GPs in the decision tool arm indicated moderate support for the tool. A minority of patients fully engaged with the mobile app-based tracking of symptoms, medication adherence and side effects.
**Conclusions** Overall, feasibility was not shown in the current study and the following modifications would be needed to attempt to overcome the limitations found: (a) inclusion of patients who have only tried one Selective Serotonin Reuptake Inhibitor, rather than two, to improve recruitment and pragmatic relevance of the study; (b) approaching community pharmacists to implement tool recommendations rather than GPs; (c) further funding to directly interface between the decision support tool and self-reported symptom app; (d) increasing the geographic reach by not requiring detailed diagnostic assessments and replacing this with supported remote self-report.
**Trial registration number** NCT03628027.

## INTRODUCTION

The last 70 years have seen the development of a wide range of antidepressants. In UK primary care, three first-line antidepressants are primarily used for the treatment of depression (fluoxetine, sertraline and citalopram), all of which are Selective Serotonin Reuptake Inhibitors.[1] However, less than one-third of patients fully recover after treatment with one of these antidepressants, meaning that many require further treatment.[2] The UK National Institute for Clinical Excellence (NICE) has further recommendations for second-line and third-line antidepressants. A lack of personalised, sequenced guidance, however, creates ambiguity for general practitioners (GPs) around the most effective next antidepressant to prescribe.[3] This is of concern, as national prescription data show frequent and rising use of certain second-line antidepressants such as venlafaxine and mirtazapine without clear decision strategies.[4] There is a clear need for further research into how structured and individualised treatment decision-making can be applied to ensure that treatment-resistant patients receive optimal antidepressant treatment.

One way to provide structured treatment guidelines is through algorithms which incorporate various patient characteristics and allocate treatments most likely to be effective (Harrison *et al*[5]). Previous research such as the Sequenced Treatment Alternatives to Relieve Depression (STAR*D) trial has applied such structured guidelines using

algorithms with success, however, the augmentation strategies used are not recommended to be carried out by GPs according to NICE guidelines.[6] Furthermore, all known trials of applying algorithms for the treatment of depression have occurred outside of the UK, which has a unique national healthcare system compared with the private healthcare settings often used in these studies. The recently published PRedDicT study applied a predictive algorithm to prescribe antidepressant treatment,[7] but was only used in the early stages of treatment, rather than for stepped treatment over the entire treatment period as in the current trial. PReDicT did not find a significant difference between use of the algorithm and treatment-as-usual in reducing depressive symptoms.

Hence, at the time of publishing the study protocol, to the authors' knowledge there was no scientifically evaluated and pragmatic stepped antidepressant decision support tool in UK primary care.[5] We confirmed this to still be the case with an updated literature search. The aim of the current study was to assess the feasibility of a future definitive randomised controlled trial to test the efficacy and cost-effectiveness of a computerised decision support tool incorporating an algorithm to advise GPs on antidepressant prescribing for patients who have not responded to first-line treatments.

### Study objectives
Our objectives were to describe (a) the recruitment of GP practices and enrolment of patients, (b) baseline patient characteristics, (c) report the prespecified feasibility outcomes and (d) provide descriptive summaries of the chosen clinical outcomes.

### METHODS
### Design
The Antidepressant Advisor Study (ADeSS) was a feasibility cluster-randomised clinical trial of a computerised decision support system for antidepressant prescribing in UK primary care. The trial was randomised at the GP practice level, where GP practices formed the clusters. At each practice, a single GP could participate at any given time. However, if a GP left the practice, a replacement GP from that practice could take their place. Most outcome measures were based on individual patient measures who were recruited to be seen by participating GPs.

### Eligibility criteria
Inclusion criteria for GP practices were: (a) up to one GP/practice participating at any time; located within one of the study's South East London areas; and (b) using EMIS electronic health record software. Inclusion criteria for patients in addition to being registered at one of the participating practices were: (a) age ≥18, (b) at least moderately severe major depressive syndrome on Patient Health Questionnaire (PHQ-9; a score of ≥15),[8] (c) no plans to change GP practice, (d) able to complete self-report scales orally or in writing, (e) no previous prescription of mirtazapine or vortioxetine, (f) evidence of early treatment resistance as defined by (i) current or recent prescription (in the last 2 months) of any of the following antidepressants listed: citalopram, fluoxetine, sertraline, escitalopram, paroxetine, venlafaxine or duloxetine, and (ii) previous prescription of at least one other antidepressant out of the same list of antidepressants.

Exclusion criteria for patients were: (a) inability to consent to the study, (b) unstable medical condition (assessed based on in-depth screening visit), (c) currently being treated by mental health specialist, (d) high suicide risk (assessed with Mini International Neuropsychiatric Interview suicidality screen),[9] (e) past diagnosis of schizophrenia or schizo-affective disorder, (f) current psychotic symptoms (three clinical screening questions validated in our previous work to exclude schizphreniform disorders),[10 11] (g) bipolar disorder on WHO Composite International Diagnostic Interview[12] at prescreening or using the Structured Clinical Interview for DSM-5[13] at screening including Bipolar Otherwise Specified categories, (h) currently at risk of being violent (assessed on in-depth screening visit), (i) drug (modified PHQ) or alcohol abuse (PHQ)[8] over the last 6 months, (j) suspected central neurological condition (eg, dementia, stroke, assessed on in-depth screening visit), (k) (planned) pregnancy or insufficient contraception in women of childbearing age (assessed on in-depth screening visit and prescreening), (l) breast feeding or within 6 months of giving birth, (m) has already been prescribed both escitalopram and sertraline.

### Recruitment of GP practices
The GP practice recruitment period was from September 2018 until March 2020, when the study had to be stopped due to the COVID-19 pandemic. During this time, 70 GP practices in Lambeth were approached as well as several other practices in South East London. Of these, 20 (29%) were recruited into the study and randomised. Of the 20 randomised practices, 10 practices (from a single Research and Development office) were withdrawn from the study shortly after randomisation and prior to training. The withdrawn practices initially expressed an interest in participating but subsequently failed to respond to all attempts to contact.

### Recruitment of patients
Patients at participating GP practices were enrolled in three stages: first, a search was conducted (via EMIS) to identify potentially eligible patients from among those registered at each GP practice. Second, patients meeting initial screening criteria were sent a letter inviting them to participate in the study and attend a prescreening assessment (conducted online or by phone). Third, patients who met eligibility criteria assessed at prescreening were then invited to attend a face-to-face screening interview where further eligibility criteria were assessed. Please refer to the trial protocol for details regarding these procedures.[5]

## Measures

### Feasibility outcomes

The primary feasibility outcome was:

1. The number and percentage of patients lost to follow-up.

   Secondary feasibility outcomes were:

2. GP adherence to the algorithm for each completed patient rated by a trial clinician (0=none of recommended steps implemented; 1=less than 50% of recommended steps implemented; 2=50% or more of recommended steps implemented; 3=100% of recommended steps implemented).
3. Average patient adherence to prescribed medications based on EMIS electronic prescribing records.
4. Adverse event (AE) and serious adverse event (SAE) rates (grade and relationship to intervention).
5. Patient adherence to GP attendance measured by % of attended GP visits out of scheduled visits on EMIS over treatment period.
6. Recruitment rates.
7. Average GP satisfaction with decision support tool (intervention arm; after GP completion of study).
8. DSM-IV Social and Occupational Functioning Assessment Scale[13] of psychosocial functioning on final visit, while modelling baseline score.
9. Maudsley Visual Analogue Mood Scale (MVAS) on final visit, while modelling baseline score.

### Primary clinical outcome measure

10. Self-rated Quick Inventory of Depressive Symptomatology sum score (QIDS-SR16[14]) at final visit, adjusting for baseline score.

### Secondary clinical outcome measures

11. Depressive symptoms were assessed by the Montgomery-Asberg Depression Rating Scale (MADRS[15]) at follow-up assessment, adjusting for baseline score, by a rater blinded to treatment allocation.
12. Clinical Global Impression (CGI) scale,[16] change between baseline and follow-up assessment assessed by a rater who was blind to treatment allocation.
13. Generalised Anxiety Disorder-7[17] scale at follow-up assessment, adjusting for baseline score.
14. Body mass index at follow-up assessment adjusting for baseline score.

### Exploratory clinical outcome measures

15. Average score for medication side effects on Frequency, Intensity, Burden of Side Effects Rating (FIBSER)[18]; self-report via mobile app.
16. Average % of adherence to prescribed antidepressant medication (self-report via mobile app).
17. Average Maudsley Modified Patient Health Questionnaire-9 (MM-PHQ-9)[19] scores in last 2 weeks (at follow-up, while modelling first 2 weeks as baseline average).

### Health economic measures

18. Service use as determined on EMIS including psychiatric referrals and referrals to study psychiatrist, as well as time to psychiatric referral; also primary care consultation rates.
19. Service use; self-reported using a modified version of the Adult Service Use Schedule.[20]
20. Quality of life using the EQ-5D-3L[21] the standard measure recommended by NICE for use in cost-effectiveness analyses

Refer to the published protocol for further details[5] regarding our assessment, as well as randomisation, blinding and sample size calculation. Methods and results pertaining to the economic evaluation can be found in online supplemental materials and are of limited interpretability due to our small sample size.

## Interventions

ADeSS was a two-arm cluster-randomised study. GP practices were allocated either to the intervention arm (herein 'Decision tool') or the control arm (herein 'treatment-as-usual'; TAU). In the Decision tool arm, patients received treatment from GPs who were using the computerised decision support tool. The tool assisted with antidepressant prescriptions and prompted GPs to review patients' medications and change them if ineffective. The algorithm and technical requirements of the tool are described in the trial protocol.[5] In the TAU arm, patients received treatment from GPs who are not assisted by the computerised decision support tool. These GPs were not aware of the treatment algorithm used in the Decision tool arm.

Patients meeting the above eligibility criteria and consenting to participate in the study attended the participating GP in their practice for treatment over 14 weeks. Patients received the intervention or control based on the arm that their GP practice was allocated to. Side effects were assessed for each week of the treatment period via the mobile app using the FIBSER[18] scale via a notification on their phone.

For patients who were not able to use the mobile app (eg, incompatible phone or other technical difficulties), weekly FIBSER scores were collected by the study team via telephone/email. Follow-up assessments took place at 15–18 weeks after the baseline interview.

## Statistical analyses

The primary feasibility outcome (#1, 'Number of patients lost to follow-up') was summarised with frequencies and percentages with exact 95% CIs (Clopper-Pearson method[22]). Other outcomes were described using appropriate summary statistics. Continuous outcomes (outcomes #3, #7–11, #13–17) were described using means and SD or medians and quartiles. Count outcomes (#6) were summarised using counts and incidence rates with exact Poisson CIs (conf.int function from the epiR package). Categorical outcomes (#2, #4–5, #12) were summarised using frequencies and percentages.

Several continuous outcomes (#8–9; clinical outcomes #13–14, #16–17, #20) were analysed using linear regression models where the dependent variable was the follow-up score and each model included (a) a dummy variable representing treatment allocation (1='Decision tool'; 0='TAU') and (b) the baseline score. To account for the clustering of patients within GP practices, SEs were adjusted using a sandwich estimator[23 24] using the *sandwich* package for R.[25] Arm differences were discerned by examining the unstandardised regression coefficients (ie, mean difference in outcome between arms). Analyses involving economic evaluation outcomes (#10, #11, #12) are described in online supplemental materials. The categorical CGI change scale outcome (#15) was summarised by cross-tabulating (frequency, percentage) with treatment arm. The intraclass correlation (ICC) for the primary clinical outcome was estimated using a random effects linear regression model, described in online supplemental materials.

For individual scales, we used published guidance on how to handle missing items. Where such guidance was not available, scales were pro-rated for individuals where 20% or less of items were missing. FIBSER scores were summarised based on item three ('Burden': 'In the past week, how much have the side effects to your medications for depression interfered with your day-to-day activities?') of the scale (see online supplemental materials for details). Inter-rater reliability for the MADRS was assessed using a two-way mixed model, absolute agreement, single measure in IBM SPSS V.15 and showed excellent inter-rater reliability (see online supplemental materials).

No participants were missing information at baseline, therefore, no imputation was carried out. Participants with missing follow-up information were excluded from regression models. No sensitivity analyses or subgroup analyses were performed. 95% CIs were treated as underpowered and not used as the basis for inferential conclusions. P values were not presented for any analyses. Analyses were conducted using R V.4.0.4.

### Patient and public involvement

The study was supported by service users which provide input to the study. We have scheduled regular meetings with our service users and one of our service user representatives has read our trial protocol publication and commented on it before submission. Due to the pandemic we have not been able to run wider public engagement workshops. We have finalised a lay summary report with our service user representatives for distributing to all participants of the study.

## RESULTS

Results for study outcomes below are ordered conceptually rather than by importance or whether they were related to feasibility or explored clinical outcomes for future trials. For our health economic results showing the absence of secondary care use in our sample (see Supplementary Results and Supplementary Table A, B, C).

### Patient enrolment and baseline characteristics
#### Number of patients enrolled per month
Each practice enrolled between 0 and 4 patients during the enrolment period, a mean of one per month (95% Poisson CI 0.59 to 1.58). Figures 1 and 2 provide a Consolidated Standards of Reporting Trials diagram describing the number of patients included at each stage of the screening process. To explore how removing the requirement for patients to have been prescribed a different antidepressant to their current one, we conducted a second, exploratory EMIS search at one of the average-sized practices participating in the trial. In the revised search 334 patients were found to be eligible, a fivefold increase over the 67 found to be eligible at the same practice and timepoint using the original search criteria.

### Baseline patient characteristics
Table 1 presents demographic characteristics of enrolled patients at baseline. A total of 18 patients completed face-to-face screening and were enrolled into the study between 9 January 2019 and 11 March 2020. Baseline clinical characteristics are presented in table 2. Data were complete at baseline. Given the small numbers, we would not wish to over-interpret baseline differences between arms. However, it is worth noting a potential imbalance in female gender (57% in Decision tool vs 91% in TAU) and median depression episode duration (2.0 in Decision tool vs 7.0 in TAU). Such variables might need to be considered as covariates for covariate restricted randomisation methods for a larger trial.

### GP-based outcomes
#### Number of GP practices recruited per month
When including all recruited GP practices, a total of 20 practices were recruited. A breakdown of GP recruitment by month is presented in Supplementary Table D. The mean number of practices recruited per month was 1.11 (95% Poisson CI 0.68 to 1.72). When including only GP practices that participated in the study (ie, excluding those that were withdrawn very shortly after randomisation), a total of 10 practices were recruited, 0.56 per month (95% Poisson CI 0.27 to 1.02).

#### GP satisfaction with tool-assisted consultation flow and outcome
GPs in the Decision tool arm were invited to complete a satisfaction survey at the end of their time in the trial. There were five GP practices in the Decision tool arm, although only four of these practices saw at least one patient and the practice not having any eligible patients also did not return a GP satisfaction questionnaire. Only 3/5 practices overall responded to the survey.

The responses from the survey are presented in Supplementary Table E. To summarise:
1. 2/5 GPs found the decision tool to be 'Possibly helpful', 1/5 'Definitely helpful' and 2/5 did not respond.

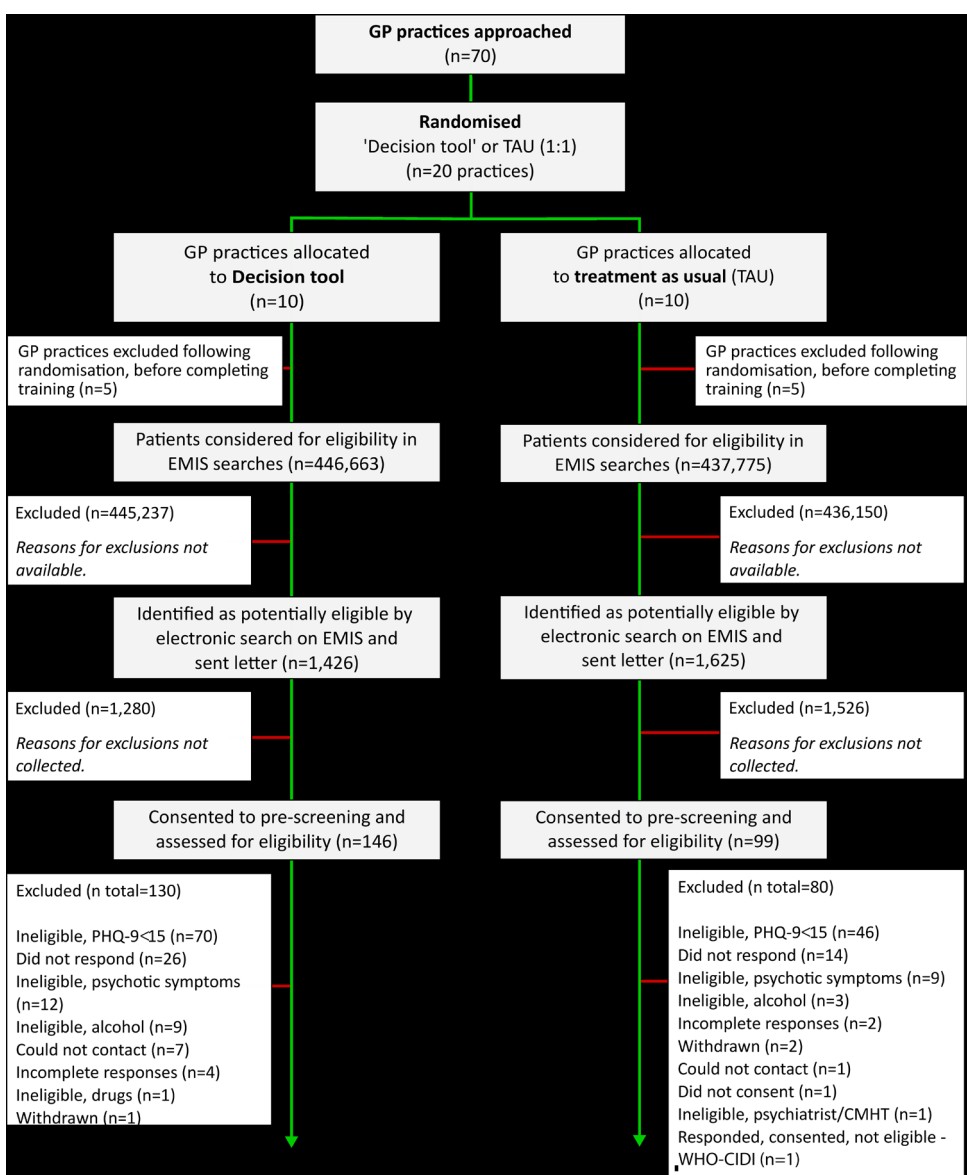

**Figure 1** Consolidated Standards of Reporting Trials (CONSORT) flow diagram. A CONSORT diagram describing the participant flow and exclusion from prescreening to follow-up for the trial. GP, general practitioner; PHQ-9, Patient Health Questionnaire-9; WHO-CIDI, WHO Composite International Diagnostic Interview.

2. 3/5 found the tool 'Slightly easy' or 'Easy' to use; 2/5 did not respond.
3. 3/5 indicated that they 'Weakly support' recommending that the EMIS Antidepressant Advisor tool be used in future clinical practice; 2/5 did not respond.

### GP adherence to the algorithm
Supplementary Table F presents, for each GP in the Decision tool arm, the number of patients in each adherence category (and percent, relative to total number of patients seen by the GP). These data present a mixed picture. While for 4/7 patients the algorithm was 'Fully implemented' by the GPs, for 3/7 patients GPs implemented 'None of the recommended steps'. It is important to note that while some GPs may have chosen to not implement the algorithm, some will have not implemented for reasons outside their control. For example, information

required for the algorithm (weekly MM-PHQ-9) was not always available and some patients did not accept proposed changes in their medication.

### Patient-based outcomes
Table 3 presents data completeness and descriptive statistics for outcomes at the follow-up interview. 'Data completeness' here refers to the number of patients for whom a follow-up score could be derived ('No. complete') compared with the number of patients attending at baseline ('No. total').

### Loss to follow-up (primary feasibility outcome)
Only 1/18 (5.6%; 95% CI 0.1 to 27.3) participants failed to attend their follow-up interview at 15/18 weeks after baseline. The single patient not attending follow-up was in the TAU arm. In total, five patients had their follow-up

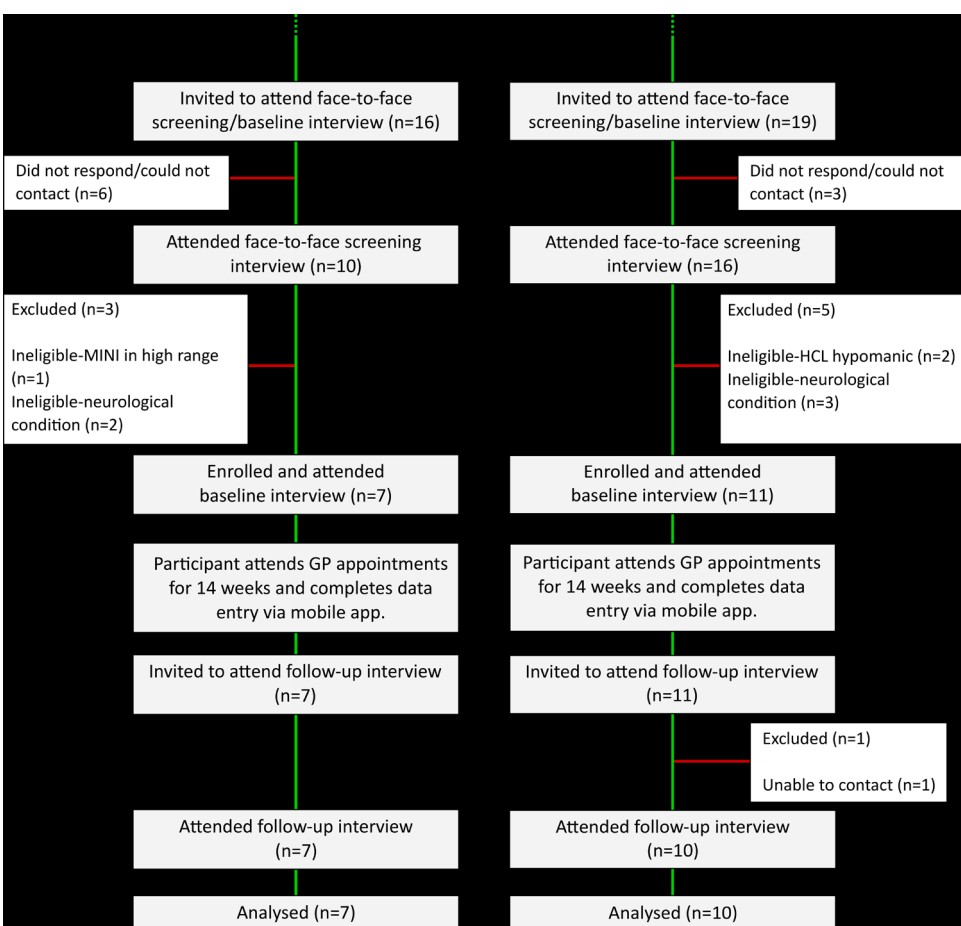

**Figure 2** Consolidated Standards of Reporting Trials (CONSORT) flow diagram continued. A CONSORT diagram describing the participant flow and exclusion from prescreening to follow-up for the trial. GP, general practitioner; MINI, Mini International Neuropsychiatric Interview; HCL, Hypomania Checklist-16.

interview outside the 15–18 weeks window because of difficulties contacting patients or scheduling the interview. This was despite offering remote video or phone consultations as an alternative to face-to-face consultations as per protocol which allowed us to continue follow-up visits throughout the pandemic.

### Adverse events

Most adverse events were attributable to expected antidepressant side effects, such as tiredness, loss of libido and nausea (see also Supplementary Table G). No SAEs were recorded during the trial. In total, there were eight mild AEs (affecting eight patients) and two moderate AEs (affecting one patient). There was an equal number of mild AEs in the Decision tool and TAU arms. The two moderate AEs were recorded for a patient in the TAU arm; no moderate AEs were recorded in the Decision tool arm.

### Score for medication side effects during follow-up

5/18 patients were not using the mobile app and their weekly FIBSER scores were collected by telephone or email. FIBSER completion rates in the first week were good (83% overall), but this fell in subsequent weeks and around 20%–30% of patients completed FIBSER

during the final weeks of the treatment period (Supplementary Table H). For 5/18 patients, no FIBSER scores were recorded in any week of the treatment period. For the remaining 13, mean scores item 3 ('Burden') were similar at 2.2 and 2.1 for the Decision tool and TAU arms, respectively. A score of 2 corresponds to the category 'Minimal interference'.

### Treatment effects for continuous clinical outcomes

Table 4 presents treatment effects and 95% CIs from linear models for continuous outcomes measured at follow-up. The treatment effects represent the mean difference in score at follow-up for patients in the Decision tool arm (compared with patients in TAU), after adjusting for the baseline score.

Given the small numbers involved, and the aims of this feasibility trial, these estimates should be treated as exploratory and not used as the basis for inferential conclusions. The CIs are wide and all included zero. We note, however, that the observed differences are in the expected direction, indicating possible improvement for patients in the Decision tool arm, compared with TAU.

**Table 1** Baseline demographic characteristics by arm

| Characteristic | Overall, N=18 | Decision tool, N=7 | TAU, N=11 |
|---|---|---|---|
| Age | | | |
| Mean (SD) | 51.4 (14.1) | 53.1 (14.2) | 50.4 (14.6) |
| Median (IQR) | 52.5 (45.2, 57.5) | 54.0 (48.5, 57.0) | 51.0 (41.5, 57.0) |
| Sex | | | |
| Male | 4 (22%) | 3 (43%) | 1 (9.1%) |
| Female | 14 (78%) | 4 (57%) | 10 (91%) |
| Other | 0 (0%) | 0 (0%) | 0 (0%) |
| Ethnicity | | | |
| Black | 2 (11%) | 0 (0%) | 2 (18%) |
| Mixed | 1 (5.6%) | 0 (0%) | 1 (9.1%) |
| Other | 2 (11%) | 0 (0%) | 2 (18%) |
| White | 13 (72%) | 7 (100%) | 6 (55%) |
| Native first language | | | |
| English | 14 (78%) | 5 (71%) | 9 (82%) |
| Non-English | 4 (22%) | 2 (29%) | 2 (18%) |
| Years of full-time education | | | |
| Mean (SD) | 13.9 (3.0) | 14.1 (3.2) | 13.7 (3.0) |
| Median (IQR) | 13.0 (11.2, 15.8) | 13.0 (12.0, 15.5) | 13.0 (11.5, 16.0) |

Baseline demographic characteristics for patients who were enrolled (N=18) in the trial (N=18 available for all measures).
TAU, treatment-as-usual.

## CGI scale

Online supplemental table I summarises the 7-category 'Clinical Global Impression' scale (CGI).[16] While inspection of the percentages might suggest that patients in the Decision tool arm were less ill, compared with TAU, these percentages are based on very small frequencies, and so we advise that these numbers be interpreted with caution.

## Patient adherence to treatment based on EMIS electronic prescribing records

Supplementary Table J presents the percentage of scheduled GP appointments that were attended by patients (including phone consultations) over the treatment period. This information was collected from EMIS records for 15/18 patients. One patient did not attend follow-up (and therefore, EMIS data were not extracted); two further patients attended their follow-up interview but EMIS data could not be extracted. Overall, most patients attended most scheduled appointments. Of patients with EMIS data (15/18), nearly 100% of scheduled appointments were attended.

## Adherence to prescribed antidepressant measured via mobile app

Uptake of the mobile app was low and some patients who initially agreed to use the app experienced technical difficulties (eg, unable to log in, missed notifications). Initial inspection of the data indicated that there were insufficient reports of daily adherence to summarise this outcome. Therefore, we report data completeness among enrolled patients. The number of doses of prescribed antidepressants could not be analysed as this data was not available from EMIS. Data completeness is presented in

**Table 2** Baseline clinical characteristics by arm

| Characteristic | Overall | Decision tool | TAU |
|---|---|---|---|
| Body mass index (BMI) | | | |
| Mean (SD) | 29.5 (8.0) | 28.9 (7.5) | 29.9 (8.7) |
| Median (IQR) | 27.8 (23.1, 35.7) | 25.8 (23.1, 34.7) | 29.3 (23.1, 36.0) |
| Age of depression onset | | | |
| Mean (SD) | 18.3 (10.0) | 18.3 (10.7) | 18.3 (10.1) |
| Median (IQR) | 15.5 (11.0, 22.2) | 14.0 (12.5, 20.5) | 17.0 (10.5, 23.5) |
| Depression episode duration | | | |
| Mean (SD) | 21.8 (53.5) | 7.6 (9.6) | 30.8 (67.6) |
| Median (IQR) | 6.5 (2.0, 11.8) | 2.0 (2.0, 11.0) | 7.0 (5.5, 11.5) |
| Number of depressive episodes | | | |
| Mean (SD) | 6.3 (6.8) | 9.6 (10.1) | 4.2 (2.1) |
| Median (IQR) | 4.5 (2.2, 6.8) | 6.0 (3.0, 12.0) | 4.0 (2.5, 6.0) |
| Illness duration | | | |
| Mean (SD) | 34.1 (18.4) | 34.9 (19.7) | 33.5 (18.4) |
| Median (IQR) | 37.5 (20.2, 46.0) | 40.0 (23.0, 42.5) | 35.0 (20.5, 47.0) |
| Number of suicide attempts | | | |
| Mean count | 0.22 | 0.29 | 0.18 |
| Poisson 95% CI | (0.06 to 0.57) | (0.03 to 1.03) | (0.02 to 0.66) |
| MINI suicidality screen total score | | | |
| Mean (SD) | 2.7 (3.3) | 2.7 (3.5) | 2.7 (3.4) |
| Median (IQR) | 1.0 (0.0, 4.8) | 1.0 (0.0, 4.5) | 1.0 (0.0, 5.5) |
| MINI suicidality screen, risk category, n (%) | | | |
| Low | 16 (89%) | 6 (86%) | 10 (91%) |
| Moderate | 2 (11%) | 1 (14%) | 1 (9.1%) |
| High | 0 (0%) | 0 (0%) | 0 (0%) |
| Maudsley staging of treatment resistance, n (%) | | | |
| Mild (3–6) | 13 (72%) | 5 (71%) | 8 (73%) |
| Moderate (7–10) | 5 (28%) | 2 (29%) | 3 (27%) |
| Severe (11–15) | 0 (0%) | 0 (0%) | 0 (0%) |
| Young Mania Rating Scale | | | |
| Mean (SD) | 2.5 (1.9) | 2.3 (1.7) | 2.6 (2.0) |
| Median (IQR) | 2.5 (1.0, 4.0) | 2.0 (1.0, 4.0) | 3.0 (1.0, 4.0) |

Baseline clinical characteristics for patients who were enrolled (N=18) in the trial (N=18 available for all measures, Decision tool group: N=7, treatment-as-usual (TAU) group: N=11).
MINI, Mini International Neuropsychiatric Interview.

online supplemental figure A. Overall, most patients did not provide regular reports of medication adherence via the mobile app. Three patients responded on 74%, 35% and 13% of days during the treatment period, respectively. All other patients responded on fewer than 8% of days.

## GP practice effect on primary clinical outcome

The ICC for QIDS-SR16 at follow-up (among 17 patients at 9 practices) was 0.07, indicating that 7.3% of variance in patient scores was attributable to differences between GP practices, after taking into account treatment allocation and baseline score. However, the 95% bootstrap CIs ranged from <0.001 to 0.76, highlighting the high degree of uncertainty in this estimate.

**Table 3** Descriptive statistics and data completeness for clinical outcomes at follow-up

| Characteristic | Overall | Decision tool | TAU |
|---|---|---|---|
| QIDS-SR16 | | | |
| No. complete/No. total (%) | 17/18 (94%) | 7/7 (100%) | 10/11 (91%) |
| Mean (SD) | 12.8 (6.3) | 10.3 (4.9) | 14.6 (6.7) |
| Median (IQR) | 15.0 (8.0, 17.0) | 9.0 (7.5, 12.0) | 16.5 (12.8, 17.0) |
| SOFAS | | | |
| No. complete/No. total (%) | 16/18 (89%) | 7/7 (100%) | 9/11 (82%) |
| Mean (SD) | 58.9 (14.5) | 66.6 (12.0) | 53.0 (13.9) |
| Median (IQR) | 59.0 (51.0, 62.8) | 61.0 (60.5, 74.0) | 55.0 (45.0, 58.0) |
| MVAS | | | |
| No. complete/No. total (%) | 16/18 (89%) | 7/7 (100%) | 9/11 (82%) |
| Mean (SD) | 92.9 (59.2) | 66.1 (38.3) | 113.8 (66.0) |
| Median (IQR) | 108.0 (39.0, 138.0) | 53.0 (37.0, 94.5) | 135.0 (106.0, 148.0) |
| MADRS | | | |
| No. complete/No. total (%) | 17/18 (94%) | 7/7 (100%) | 10/11 (91%) |
| Mean (SD) | 21.9 (10.4) | 16.7 (7.6) | 25.6 (10.8) |
| Median (IQR) | 25.0 (12.0, 30.0) | 13.0 (11.5, 20.0) | 28.0 (23.0, 31.0) |
| GAD-7 | | | |
| No. complete/No. total (%) | 17/18 (94%) | 7/7 (100%) | 10/11 (91%) |
| Mean (SD) | 8.0 (6.4) | 6.3 (5.3) | 9.2 (7.1) |
| Median (IQR) | 6.0 (4.0, 13.0) | 5.8 (3.5, 6.5) | 8.0 (4.5, 15.2) |
| BMI | | | |
| No. complete/No. total (%) | 15/18 (83%) | 7/7 (100%) | 8/11 (73%) |
| Mean (SD) | 29.8 (8.2) | 28.0 (7.4) | 31.5 (9.0) |
| Median (IQR) | 32.7 (22.5, 35.3) | 24.6 (21.4, 35.0) | 33.2 (25.6, 35.8) |
| Maudsley Modified PHQ-9 | | | |
| No. complete/No. total (%) | 17/18 (94%) | 7/7 (100%) | 10/11 (91%) |
| Mean (SD) | 12.0 (7.8) | 8.9 (5.7) | 14.2 (8.5) |
| Median (IQR) | 13.0 (5.0, 18.0) | 8.0 (4.5, 12.0) | 16.5 (8.5, 20.8) |

Descriptive statistics for clinical outcome measures at follow-up: overall N=18 enrolled, Decision tool group: N=7, treatment-as-usual (TAU) group: N=11.
We lost one patient to follow-up in the TAU group.
BMI, body mass index; GAD-7, Generalised Anxiety Disorder-7 scale; MADRS, Montgomery-Asberg Depression Rating Scale; MVAS, Maudsley Visual Analogue Scale; PHQ-9, Patient Health Questionnaire-9; QIDS-SR16, Self-rated Quick Inventory of Depressive Symptomatology sum score; SOFAS, Social and Occupational Functioning Assessment Scale.

## DISCUSSION

This trial investigated the feasibility of a cluster-randomised design to study a novel computerised Anti-depressant Advisor tool for UK primary care which was developed as part of this study. While the loss to follow-up rate (the primary feasibility outcome) was very low and the software implementation of our algorithm was successful and raised no safety issues, both GP practice and patient recruitment were slower than anticipated, resulting in a much smaller sample size than planned. The GP practice recruitment strategy was partially successful, in that GPs were interested in the study but the recruitment rate was slow, largely due to our restriction of only being able to recruit in South London. A national recruitment strategy with remote consultations and/or online self-assessment would have greatly increased our speed of practice recruitment. Most practices were recruited via Clinical Research Network staff, who assisted in advertising to GP practices and setting up training for recruited GPs. This is similar to findings reported by the STAR*D trial that sites where clinical research coordinators played a key role were more likely to be enrolled into the study.[26] As in the STAR*D trial, ADeSS benefited from having clinical research nurses, whose national involvement would be crucial for the success of a future larger trial.

Our patient recruitment strategy was successful in that an expected percentage of patients expressed their interest in taking part in the study (8% of those

| Outcome | Direction | N avail./N total* | Decision vs TAU mean difference β† | (95% CI)‡ |
|---|---|---|---|---|
| SOFAS§ | Higher scores indicate higher level of social functioning | 16/18 | 9.8 | (−3.2 to 22.7) |
| MVAS¶ | Higher scores indicate greater symptom severity | 16/18 | −40.3 | (−96.9 to 16.3) |
| QIDS-SR16 | Higher scores indicate greater symptom severity | 17/18 | −0.2 | (−6.9 to 6.5) |
| MADRS | Higher scores indicate greater symptom severity | 17/18 | −8.1 | (−19.3 to 3.0) |
| GAD-7 | Higher scores indicate greater symptom severity | 17/18 | −1.9 | (−7.9 to 4.2) |
| BMI** | Higher scores indicate higher body mass index (kg/m²) | 15/18 | −0.8 | (−3.3 to 1.8) |
| MM-PHQ-9 | Higher scores indicate greater symptom severity | 17/18 | −3.9 | (−10.6 to 2.8) |

**Table 4** Treatment effects for continuous clinical outcomes at follow-up from linear models

*The number of enrolled patients with non-missing information at follow-up. The maximum value here is 17/18, since one patient did not attend their follow-up interview.
†Treatment effect represents the mean difference in score at follow-up for patients in the Decision tool arm compared with patients in TAU, after adjusting for baseline score. Negative scores indicate a lower score at follow-up in the Decision support arm relative to the TAU group.
‡95% CIs were estimated using a robust sandwich estimator to account for clustering of patients within general practitioner practices.
§SOFAS is a clinician-derived score (ie, not based on a sum of individual items). Therefore, no imputation of missing item-level data was performed.
¶One patient completed no items in the MVAS.
**Two patients were missing information on weight at the follow-up interview.
BMI, body mass index; GAD-7, Generalised Anxiety Disorder-7 scale; MADRS, Montgomery-Asberg Depression Rating Scale; MM-PHQ-9, Maudsley Modified Patient Health Questionnaire-9; MVAS, Maudsley Visual Analogue Scale; QIDS-SR16, Self-rated Quick Inventory of Depressive Symptomatology sum score; SOFAS, Social and Occupational Functioning Assessment Scale; TAU, treatment-as-usual.

contacted). Patients' interest in the study and perception of it as worthwhile was also supported by the low loss-to-follow-up rate, as well as informal feedback. However, the number of patients eligible for the study limited the recruitment pace. This limitation was apparent both during searches in EMIS electronic records (0.3% of patients found to be eligible) and from eligibility of patients at prescreening (53% eligible). A large factor in limiting the pool of eligible patients in EMIS was the criterion to have previously taken an antidepressant different to their current one. Indeed, an exploratory EMIS eligibility search showed that, when the requirement for patients to have taken a previous antidepressant different to their current antidepressant was removed, the number of eligible patients increased fivefold. Additionally, the main reason for exclusion at prescreening was a PHQ-9 score of at least 15, comprising 55% of exclusions and one may question whether one may use a lower cut-off score in future.

GP satisfaction with the advisor tool was moderate, but due to our low sample size, it is difficult to draw firm conclusions around the usability and acceptance of the tool in primary care. GP satisfaction is a crucial criterion for successful implementation of the advisor tool in a definitive trial, therefore additional feedback would be required before progressing. Independently of GPs' priorities, given the large treatment gaps for depression confirmed in a recent paper, there was a consensus for introducing decision support systems as one of a set of recommendations to improve the fact that only a minority of patients with depression receive guideline-based care.[27] Similarly to the PReDicT study,[7] our study also showed that even when prompted to change treatment, this often is not adhered to by GPs and other barriers, particularly resource implications such as a shortage of follow-up appointments required for medication changes need to be addressed for algorithms to be implemented.

The mobile app enabled regular reporting of MM-PHQ-9 scores to GPs for use in treatment. However, almost half of patients did not use the app at all and, among those who did use it, only around half completed their weekly MM-PHQ-9 scores. Collecting MM-PHQ-9 and FIBSER scores via phone was very time-consuming and would not be scalable to a larger study. There are several potential reasons for the lack of use of the app. The app was not available to download directly from the Apple App Store and had to be downloaded via another app as a test version, which introduced additional complexities. Additionally, patients regularly reported technical errors where the app stopped working and needed to be updated or re-downloaded.

One limitation of our study was the lack of a more in-depth qualitative evaluation of user perspectives on the decision support system as well as the mobile app and future trials could embed this into further optimisation of their design. One of the main limitations of the trial was its geographical limitation to South London and disruption due to the COVID-19 pandemic. A future trial,

should expand the geographical reach for recruitment to increase the sample size and to employ remote assessments, without the need for in depth in-person diagnostic assessments. Another limitation was the reliance of the study on GPs, who are struggling with their workloads, to run the decision tool. It may be beneficial for future studies to use other health practitioners such as pharmacists, who could share the burden of using the tool with GPs. Indeed, the Royal Pharmaceutical Society has emphasised the important role pharmacists can play in providing treatment and improving patient outcomes as part of Primary Care Networks.[28] The clinical pharmacists in general practice programme has already demonstrated the utility of incorporating pharmacists into general practice for easing workloads on GPs. Hence, it would be a progressive step for future studies to design an advisor tool around a more collaborative primary care, rather than solely aimed at GPs.

Low uptake of the mobile app meant GPs often lacked the necessary information to run the advisor tool. A recent systematic review and meta-analysis found that trials of apps for depressive symptoms which incorporated human feedback had lower dropout rates.[29] The app and advisor tool were designed to work in sync, so that GPs could provide feedback and treatment based on regular app-reported symptoms. However, low app use combined with GP lack of adherence to the advisor tool may have resulted in the breakdown of this process, leading to patients not experiencing the human feedback required to maintain high app engagement. Future studies should prioritise the link between the advisor tool and app, both in the interest of GPs receiving the necessary app-reported symptoms to run the tool as well as patients receiving feedback on their symptoms. Additionally, future studies may wish to consider other, more reliable formats of data collection such as existing online survey tools, as well as requiring a lower frequency of data entry such as weekly rather than daily, if a study is to take place over a few months.

The main implications of this feasibility trial are that while computerised decision support tools for antidepressant prescribing are technically feasible and well placed to address important treatment gaps in UK primary care, their implementation is unlikely to be feasible by solely relying on GPs without additional case management, for example, by community pharmacists or prescribing nurses. Our study highlights that many patients remained on one antidepressant even if they had not sufficiently responded and that switching even to a second alternative was often not implemented.

**Author affiliations**
[1]Centre for Affective Disorders, Department of Psychological Medicine, Institute of Psychiatry Psychology and Neuroscience, King's College London, London, UK
[2]Department of Biostatistics and Health Informatics, Institute of Psychiatry, Psychology & Neuroscience, King's College London, London, UK
[3]National Service for Affective Disorders, South London and Maudsley Mental Health NHS Trust, London, UK
[4]Department of Population Health Sciences, King's College London, London, UK
[5]Department of Health Services & Population Research, Institute of Psychiatry, Psychology & Neuroscience, King's College London, London, UK

**Acknowledgements** The authors wish to thank Professor Emeritus André Tylee and Dr Daniel Dietch for their critical contribution to the initial planning of this study and the following members of the Trial Steering Committee who have dedicated their time and provided valuable advice: Professor Glyn Lewis, Dr Victoria Cornelius, Dr Sarah Markham and Mrs Evelyn London. We are also grateful to EMIS PLC with whom we have designed the software implementation of our Antidepressant Advisor decision support tool and to Alloc Modulo LTD with whom we have developed the accompanying MooDoC mobile app.

**Contributors** PH drafted the manuscript. EC and KG conducted the statistical analysis and wrote the trial report on which this manuscript is based. RZ finalised the draft and acts as guarantor. EC, KG, AY, MA, DF, SD, BMB all commented significantly on drafts of the manuscript. BMB provided the economic evaluation. PH, DF and SD collected data for the study. AHY, MA and RZ provided oversight on the study procedures and delivery. RZ, AHY, KG and MA designed the study.

**Funding** This paper represents independent research funded by the National Institute for Health Research (NIHR) research for patient benefit scheme (PB-PG-0416-20039) and independent research part funded (KG, EC, RZ, AY) by the National Institute for Health Research (NIHR) Biomedical Research Centre at South London and Maudsley NHS Foundation Trust and King's College London and the Applied Research Collaboration South London (NIHR ARC South London) at King's College London. The views expressed are those of the authors and not necessarily those of the NHS, the NIHR or the Department of Health and Social Care. DF's PhD is funded by the Medical Research Council Doctoral Training Partnership (project reference: 2064430). The authors acknowledge the support provided to the study by the South London Clinical Research Network and sponsorship by Lambeth CCG.

**Competing interests** AHY is a consultant to Johnson & Johnson and Livanova. AHY has given paid lectures and sat on advisory boards for the following companies with drugs used in affective and related disorders: Astrazenaca, Eli Lilly, Lundbeck, Sunovion, Servier, Livanova, Janssen, Allegan, Bionomics, Sumitomo Dainippon Pharma. AHY has received honoraria for attending advisory boards and presenting talks at meetings organised by LivaNova. AHY is the Principal Investigator of the following studies: Restore-Life VNS registry study funded by LivaNova, ESKETINTRD3004: 'An Open-label, Long-term, Safety and Efficacy Study of Intranasal Esketamine in Treatment-resistant Depression', 'The Effects of Psilocybin on Cognitive Function in Healthy Participants' and 'The Safety and Efficacy of Psilocybin in Participants with Treatment-Resistant Depression (P-TRD)'. AHY has received grant funding (past and present) from the following: NIMH (USA); CIHR (Canada); NARSAD (USA); Stanley Medical Research Institute (USA); MRC (UK); Wellcome Trust (UK); Royal College of Physicians (Edin); BMA (UK); UBC-VGH Foundation (Canada); WEDC (Canada); CCS Depression Research Fund (Canada); MSFHR (Canada); NIHR (UK); Janssen (UK). RZ is a private psychiatrist service provider at The London Depression Institute and co-investigator on a Livanova-funded observational study of Vagus Nerve Stimulation for Depression. RZ has received honoraria for talks at medical symposia sponsored by Lundbeck as well as Janssen. He has collaborated with EMIS PLC for this study and advises Depsee Ltd. He is affiliated with the D'Or Institute of Research and Education, Rio de Janeiro and advises the Scients Institute, USA. KG reports grants from NIHR, Stroke association, National Institutes of Health (USA) and Juvenile Diabetes Research Foundation (USA) during the conduct of the study. EC reports personal fees from NIHR during the conduct of the study. BMB reports grants from NIHR, National Institutes of Health (USA) and Guys and St. Thomas' Foundation during the conduct of the study. The other authors report no competing interests.

**Patient and public involvement** Patients and/or the public were involved in the design, or conduct, or reporting, or dissemination plans of this research. Refer to the Methods section for further details.

**Patient consent for publication** Not applicable.

**Ethics approval** This study involves human participants and was approved by London - Camberwell St Giles Research Ethics Committee, reference number: 17/LO/2074. Participants gave informed consent to participate in the study before taking part.

**Provenance and peer review** Not commissioned; externally peer reviewed.

**Data availability statement** Data are available upon reasonable request. We have not obtained consent for sharing pseudonymised data and will therefore only be able to share fully anonymised data such as scores on standardised instruments via

the King's Open Research Data System (https://kcl.figshare.com/), but not clinical history details.

**ORCID iDs**
Phillippa Harrison http://orcid.org/0000-0002-5039-7822
Ewan Carr http://orcid.org/0000-0002-1146-4922
Mark Ashworth http://orcid.org/0000-0001-6514-9904
Roland Zahn http://orcid.org/0000-0002-8447-1453

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
