## [Reviewer comments · BMJ Open]

ARTICLE DETAILS

TITLE (PROVISIONAL)	The Antidepressant Advisor (ADeSS), a decision support system for antidepressant treatment for depression in UK primary care: a feasibility study
AUTHORS	Harrison, Phillippa; Carr, Ewan; Goldsmith, Kimberley; Young, Allan; Ashworth, Mark; Fennema, Diede; Duan, Suqian; Barrett, Barbara; Zahn, Roland

VERSION 1 – REVIEW

REVIEWER	Ligthart, Suzanne
REVIEW RETURNED	06-Jul-2022

GENERAL COMMENTS	The topic and objectives of this study are relevant, since antidepressant care is often problematic. The decision aid captured my attention, for treatment of depression is often challenging. I do have some questions/concerns about the methods and conclusions drawn. First is the relevance to daily primary care practice. I came to the conclusion that the helpfulness of the computerized decision-aid is insufficient in its current form. The authors recognize that the selection criteria were too strict, which I agree with. But is it not clear if adjustment would improve the results/would the algorithm be useful in this case? Would GP's be helped? The authors suggest that community pharmacists could play a role in the further implementation. This does not take into account the problems of which patients to select, nor does it address which problem this would solve. Also, it does not answer the question if this decision-aid could be helpful for patients and professionals/what problem it solves to them. The authors have done several analyses, as planned in the protocol. However, due to the small number of participants there are wide confidence intervals, and results should be interpreted with caution, as the authors state. I could imagine presenting less numbers/analyses for this reason, and more information about the process, so that future initiatives can benefit optimally from the lessons learned from this study. I would be very curious for a more in depth analysis of the pro's and con's of this pilot: what aid could help GP's in daily practice? The app did not work properly, but if it was: would patients and GP's consider this helpful? In what way? And do GPs support the conclusion of the authors that community pharmacist could/should play a role in using the decision aid? Overall, having used the decision aid: do patients and GP's have wishes/ideas for what would really help them in clinical practice? This asks for qualitative analyses / a more in-depth process evaluation. I would want to know this, before putting further funding in the interface and/or increasing geographical need, as is advised in the
--

	manuscript and abstract. Overall, but this is no subject of this manuscript, I was in doubt about what the algorithm could bring and how evidence based its advices are. For example: to my knowledge Vortioxetine has no clear benefits above other SSRI's, and I could not find any proof of this (see for instance Cochrane sysrevs: doi: 10.1002/14651858.CD011520.pub2 / DOI: 10.1002/14651858.CD013674.pub2). In the national guidelines here, there is no advice to use this antidepressant in first-line care, but maybe this is different in the UK. If implementing a time-consuming decision-aid tool for GPs and patients, the potential benefits should be very clear. For example for smoking cessation or losing weight, benefits are clear, but still succesfull decision-aids not available. I am in doubt about the potential/theoretical benefits of this decision aid. Also, it could possibly put (too?) much emphasis on antidepressant/medication use as The intervention, while other interventions with similar effect rates and less side effects are available. Minor comments: Strengths and Weaknesses, P3: I doubt this strength due to the small number of practices and patients: A representative sample of participants was recruited from several Clinical Commissioning Groups across South London P12 L21: I think this should be in the discussion section, not results ("To explore how removing the requirement for patients to have been prescribed a different antidepressant to their current one, we conducted a second, exploratory EMIS search at one of the average-sized practices participating in the trial. In the revised search 334 patients were found to be eligible, a five-fold ") In the discussion I would like to read a reflection on the results of this study and PreDICT (introduction) From the supplementary files I see that 4 practices were included in the decision-aid group. It then seems logical that the 5th GP did not answer the questions, since he/she did not use the decision aid. I think it should be clear that the 7 included patients were recruited bij 4 practices(2 per practice and 1 of 1). P22 Discussion L44: "Patients' interest in the study and perception of it as worthwhile was also supported by the low loss-to-follow-up rate, as well as qualitative feedback" -> how was this concluded/ evaluated, what qualitative data was available? The final comment is a useful one. It's not a conclusion of the current study, but does ask for further evaluation. (' Our study highlights that many patients remained on one antidepressant even if they had not sufficiently responded and that switching even to a second alternative was often not implemented')
--	--

VERSION 1 – AUTHOR RESPONSE

Point-by-point response to comments

Reviewer: 1 - Ms. Suzanne Ligthart

The topic and objectives of this study are relevant, since antidepressant care is often problematic. The decision aid captured my attention, for treatment of depression is often challenging.

Response: Thank you for this positive feedback.

I do have some questions/concerns about the methods and conclusions drawn.

First is the relevance to daily primary care practice. I came to the conclusion that the helpfulness of the computerized decision-aid is insufficient in its current form. The authors recognize that the selection criteria were too strict, which I agree with. But is not clear if adjustment would improve the results/would the algorithm be useful in this case? Would GP's be helped? The authors suggest that community pharmacists could play a role in the further implementation. This does not take into account the problems of which patients to select, nor does it address which problem this would solve. Also, it does not answer the question if this decision-aid could be helpful for patients and professionals/what problem it solves to then.

Response: Thank you for reflecting on the usefulness of the algorithm itself. The rationale for the design of the algorithm is described in more detail in our protocol paper, published in BMJ open <https://bmjopen.bmj.com/content/10/5/e035905>. The design of the algorithm itself is based on careful considerations of efficacy, effects and side effects and would certainly address treatment gaps in primary care, also identified in our study, namely that people remain on ineffective medications long-term without changing them. We have now made this clearer by adding a sentence to the discussion and referring to a new paper on treatment gaps (see below). Having said this, the current report is about the feasibility of the trial design and not about the design of the algorithm which is largely based on guidelines and previous evidence as well as clinical experience of the investigators who have designed it.

“Independently of GPs’ priorities, given the large treatment gaps for depression confirmed in a recent paper, there was a consensus for introducing decision support systems as one of a set of recommendations to improve the fact that only a minority of patients with depression receive guideline-based care²⁸.”

The authors have done several analysis, as planned in the protocol. However, due to the small number of participants there are wide confidence intervals, and results should be interpreted with

caution, as the authors state. I could imagine presenting less numbers/analyses for this reason, and more information about the process, so that future initiatives can benefit optimally from the lessons learned from this study.

Response: Indeed as the reviewer notes, many of the results are of limited interest given the small sample size, but the CONSORT guidelines nevertheless compel us to report all the pre-registered outcomes which we have tried to compress as much as possible.

I would be very curious for a more in depth analysis of the pro's and con's of this pilot: what aid could help GP's in daily practice? The app did not work properly, but if it was: would patients and GP's consider this helpful? In what way? And do GPs support the conclusion of the authors that community pharmacist could/should play a role in using the decision aid? Overall, having used the decision aid: do patients and GP's have wishes/ideas for what would really help them in clinical practice? This asks for qualitative analyses / a more in-depth process evaluation. I would want to know this, before putting further funding in the interface and/or increasing geographical need, as is advised in the manuscript and abstract.

Response: We agree with the reviewer that it would have been very valuable to add a qualitative evaluation to our outcome measures, unfortunately we have not carried out focus groups or qualitative interviews and therefore have to rely on the responses given on our questionnaires. We have added this as a limitation to our discussion: “*One limitation of our study was the lack of a more in-depth qualitative evaluation of user perspectives on the decision support system as well as the mobile app and future trials could embed this into further optimisation of their design.*” Regarding community pharmacists, these have been rolled out in UK primary care now but were not available at the time of the study. We have cited a paper which supported this. Community pharmacists are in charge of managing prescribing in many GP practices in the UK now, so it makes sense to involve them.

Overall, but this is no subject of this manuscript, I was in doubt about what the algorithm could bring and how evidence based its advices are. For example: to my knowledge Vortioxetine has no clear benefits above other SSRI's, and I could not find any proof of this (see for instance Cochrane sysrevs: doi:10.1002/14651858.CD011520.pub2 / DOI: 10.1002/14651858.CD013674.pub2). In the national

guidelines here, there is no advice to use this antidepressant in first-line care, but maybe this is different in the UK.

Response: The second DOI link pertains to children and adolescents which is not relevant for our study, as our algorithm is only designed for adults. Thank you for pointing to the Cochrane review by Koesters et al., 2017 on Vortioxetine in adult depression (the first DOI link), which comes to a different conclusion on the basis of the same evidence as the systematic review of the evidence undertaken by our National Institute of Clinical Excellence, which has not changed the view of the UK National Institute for Clinical Excellence that Vortioxetine should be considered in adults with more severe depression (i.e. PHQ of 15 as in our study or above) who have not responded to two previous antidepressants which was recently updated in 2022 (<https://www.nice.org.uk/guidance/ng222>) to confirm an earlier technology appraisal that underpinned our algorithm design and was cited in the supplement [bmjopen-2019-035905supp001.pdf] of our protocol paper, where we outline in more detail the evidence and rationale underpinning our algorithm design and although this is of course open for debate, we do not think we have enough space to do this in the current paper, particularly because the trial feasibility was unaffected by the choice of vortioxetine more specifically.

If implementing a time-consuming decision-aid tool for GPs and patients, the potential benefits should be very clear. For example for smoking cessation or losing weight, benefits are clear, but still successful decision-aids not available. I am in doubt about the potential/theoretical benefits of this decision aid. Also, it could possibly put (too?) much emphasis on antidepressant/medication use as The intervention, while other interventions with similar effect rates and less side effects are available.

Response: The reviewer raises valid points which we have addressed in our protocol paper, where the algorithm was described in more detail. For depression, NICE recommends physical exercise as well as psychological treatments and our algorithm reminds GPs about this initially as stated in our protocol paper, but for more severe depression these are not recommended as monotherapies unless this is the explicit wish of patients. One also needs to consider that our algorithm is only triggered for patients who have already made a shared decision with their GPs to try medications and who have not responded to them, so the aim is not to discourage non-pharmacological treatments or to start more people on pharmacological

treatments, but to encourage GPs following NICE guidelines by encouraging patients to switch to other antidepressants if their current antidepressant has not worked.

Minor comments:

Strengths and Weaknesses, P3: I doubt this strength due to the small number of practices and patients: A representative sample of participants was recruited from several Clinical Commissioning Groups across South London

Response: Thank you, we have deleted the statement.

P12 L21: I think this should be in the discussion section, not results ("To explore how removing the requirement for patients to have been prescribed a different antidepressant to their current one, we conducted a second, exploratory EMIS search at one of the average-sized practices participating in the trial. In the revised search 334 patients were found to be eligible, a five-fold ")

Response: We have summarized this result in the discussion as a five-fold increase in eligible participants, but it would be uncommon to report details of an analysis in the discussion section and we think it would disrupt the flow of the discussion.

In the discussion I would like to read a reflection on the results of this study and PreDICT (introduction).

Response: Thank you, we have added the following to the discussion: *"Similarly to the PReDICT study⁶, our study also showed that even when prompted to change treatment, this often is not adhered to by GPs and other barriers, particularly resource implications such as a shortage of follow-up appointments required for medication changes need to be addressed for algorithms to be implemented."*

From the supplementary files I see that 4 practices were included in the decision-aid group. It then seems logical that the 5th GP did not answer the questions, since he/she did not use the decision aid. I think it should be clear that the 7 included patients were recruited by 4 practices (2 per practice and 1 of 1).

Response: Thank you for carefully looking at our supplement. This was indeed misleading, we have provided more detail in the results section now “There were five GP practices in the ‘Decision tool’ arm, although only four of these practices saw at least one patient and the practice not having any eligible patients also did not return a GP satisfaction questionnaire. Only 3/5 practices overall responded to the survey.”

P22 Discussion L44: "Patients' interest in the study and perception of it as worthwhile was also supported by the low loss-to-follow-up rate, as well as qualitative feedback" -> how was this concluded/ evaluated, what qualitative data was available?

Response: Thank you, the term “qualitative” was meant broadly, but is misleading indeed in that it implies a formal qualitative element to our study. We did not collect qualitative data formally, but we informally asked all our participants for informal feedback during the follow-up session. We have therefore reformulated: “as well as *informal* feedback”

The final comment is a useful one. It's not a conclusion of the current study, but does ask for further evaluation. (' Our study highlights that many patients remained on one antidepressant even if they had not sufficiently responded and that switching even to a second alternative was often not implemented')

Response: Thank you.

VERSION 2 – REVIEW

REVIEWER	Ligthart, Suzanne
REVIEW RETURNED	26-Nov-2022

GENERAL COMMENTS	Thanks to the reviewers for their answers. I feel the revised version has improved and the answers of the authors are satisfactory. Although the research did not yield the results that were expected/hoped for, they do point out the difficulties which led to these results very clearly. This is an effort to reduce treatment inertia, and we have to look for alternative ways like this to support clinicians towards optimal treatment strategies for patients. For instance by a decision support tool from the start of the 1st treatment (in my opinion; monitoring for desired effect and if not: advice) and a role for community pharmacists (which I would like to learn more about, but not in this paper).
--